# Investigation of the Viscoelastic Behavior Variation of Glass Mat Thermoplastics (GMT) in Compression Molding

**DOI:** 10.3390/polym11020335

**Published:** 2019-02-14

**Authors:** Chao-Tsai Huang, Ling-Jue Chen, Tse-Yu Chien

**Affiliations:** Department of Chemical and Materials Engineering, Tamkang University, No. 151, Yingzhuan Rd., Tamsui Dist., New Taipei City 25137, Taiwan; fo54700063@gmail.com (L.-J.C.), michale221762@gmail.com (T.-Y.C.)

**Keywords:** compression molding, glass mat thermoplastics (GMT), viscoelasticity, fiber–polymer matrix separation

## Abstract

Compression molding is a lightweight technology that allows to preserve fiber length and retain better mechanical properties compared to injection molding. In compression molding development, a suitable material such as glass fiber mat thermoplastics (GMT) is often used. However, because of the complicated micro-structure of the fibers and the fiber–resin matrix interactions, it is still quite challenging to understand the mechanism of compression molding and it is very difficult to obtain a uniformly compressed GMT product. In this study, we propose a method to measure the rheological properties of GMT through a compression system. Specifically, we utilized a compression molding system to record the relation between the loading force and the displacement. This quantitative information was used to estimate the power-law index and viscoelastic parameters and predict viscosity. Moreover, the estimated viscoelastic parameters of GMT were implemented into Moldex3D to evaluate the flow behavior under compression. The results showed that the trend of the loading force variation was consistent in numerical simulation and experiments. However, at the final stage of compression molding, the experimental loading force was much higher than that estimated by simulation. To find out the mechanism causing this deviation, a series of studies were performed. Through TGA measurement, we found that the fiber content of the center portion of the compressed part increased from 63% to 85% during compression. This was expected, as a result of the fiber–polymer matrix separation effect. This fiber–polymer matrix separation effect influenced the power-law index and rheological parameters of GMT, making them fluctuate. Specifically, the power-law index changed from 1.0 to 0.62. These internal changes of the rheological properties further induced a much higher loading force in the real experimental GMT system. We further verified the rheological properties variation using pure polyamide (PA) and found that since there is no fiber–polymer matrix interactions the power-law index and curve-fitting rheological parameters were almost constant. The mechanism causing the deviation was therefore validated.

## 1. Introduction

Because of the excellent strengthening obtained in plastics, fiber has received a great attention in lightweight technology development in recent years. However, one of the crucial problems is the breakage of long fibers which reduces the impact strength significantly during product manufacturing [1,2,3,4]. To overcome fiber breakage, many researchers suggested to use glass mat thermoplastics (GMT) through compression molding because it is a softer processing method [5,6,7,8]. However, using compression molding poses several problems, such as how to specify the types of charges and related amounts, how to carry on processing from charging to finished products, how to establish the contents of internal polymer matrix and fiber, and so on. Besides, some defects will occur, such as short shot, unbalanced flow, warpage parts, cracking, etc. Many researchers have been involved into overcoming these challenges. Kotsikos et al. [9] applied an axisymmetric squeeze testing between two parallel round plates to study the flow behavior of GMT. They found the loading (force) increased exponentially as the displacement increased. Dweib and Bradaigh [10,11,12] tried to measure the viscosity behavior under squeeze flow and model their data. They found the viscosity of GMT presents a power law feature. Moreover, Tornqvist et al. [13] tried to optimize the conditions for GMT isothermally, but this is not so easy to perform in actual applications in the industry. Furthermore, many researchers focused on the rheological modelling and characterization of fiber-reinforced plastics. Sherwood and Durban [14] applied a cylinder with two parallel plates to obtain a squeeze flow with radius R >> height h. They found their method is suitable for disc-typed specimens, but the interpretation of the results is not easy. Chan and Baird [15] tried to determine the viscosity of highly viscous filled thermoplastics using the power law and the Bingham and Herschel–Bulkley models. They found the Herschel–Bulkley model provided the best results. Colin Servais et al. [16] discovered the compression flow behavior of short-to-long fiber-reinforced thermoplastics. They found the compression behaviors of short fibers and long fibers are qualitatively different. They suggested that for short-fiber systems, the flow profile is a combination of shear-like flow in the core and Coulomb friction at the plates walls, and that the profile becomes more plug-like as the fiber length is increased. Indeed, the model for squeeze flow is still under-developed.

On the other hand, to handle the problems and challenges described earlier, researchers conventionally apply a trial-and-error method which, however, is not effective. Researchers would like to reduce the problems using computer-aided engineering (CAE) technology. Unfortunately, for GMT, CAE is not mature to perform compression molding simulations. One of the key questions is that the rheology properties (especially viscoelasticity) of GMT have not been characterized comprehensively. In general, the viscoelastic property is measured by capillary rheometer or cone-and-plate rheometer for common thermoplastics, but these two methods are not suitable for GMT [15]. In fact, the channel of the capillary rheometer will be frequently blocked by GMT because of its high fiber content. Moreover, fiber–matrix separation is another challenge for GMT in the cone-and-plate or in the two-parallel plate compression rheometer [17].

In this paper, we investigated the viscoelastic property of GMT through experimental and numerical studies. A compression system to conduct the rheological features through a compression flow using Instron machine is proposed. Furthermore, the measured compression data are transferred into the viscosity data based on literature’s models [10,11,12] to obtain the required information for further compression process simulation. Moreover, the viscosity data are integrated into CAE software (Moldex3D Compression module) to perform the compression molding processing numerically. At the same time, a real compression molding trial is performed. The rheological behavior of GMT is investigated by comparison of the experimental study and the numerical simulation. This paper is organized as follows. Section 2 describes the theory and assumption. The investigation method and related information are presented in Section 2. In Section 3, results and discussion are described. Finally, conclusions are outlined in Section 4.

## 2. Investigation Methods and Related Procedures

To study the viscoelastic behavior of GMT in compression molding, an experimental study and a numerical simulation were utilized. The details are as follows.

### 2.1. Theory and Model

The schematic for the compression system is described in Figure 1. GMT was charged into the central region before compression. During compression, GMT was squeezed out until reaching the desired thickness. In addition, it was ensured that the expansion along the radius was equal across the sample to obtain uniform thickness and loading force.

In this process, the non-isothermal 3D flow motion can be mathematically described by the following conservation laws for mass, momentum, and energy:(1)$$\frac{\partial \rho }{\partial t}+\nabla \cdot (\rho u)=0$$
(2)$$\rho (\frac{\partial u}{\partial t}+(u-u_{\mathrm{compression}})\cdot \nabla u))=-\nabla p+\nabla \cdot \tau$$
(3)$$\rho C_{P}\left(\frac{\partial T}{\partial t}+(u-u_{\mathrm{compression}})\cdot \nabla T\right)=\nabla \left(K\nabla T\right)$$
(4)$$\frac{\partial u}{\partial t}=\nabla \cdot (u_{\mathrm{compression}})$$ where *u* is the velocity vector of the fluid, *u*_compression_ is the velocity vector of the upper mold, *p* is the pressure, *T* is the temperature, *t* is compression time, **τ** is the stress tensor, *ρ*, η, *Cp*, and K represent density, viscosity, heat capacity, and thermal conductivity of the plastic material, respectively. Through the whole analysis, the polymer melt was assumed to behave as a Generalized Newtonian Fluid (GNF) with compressibility.

During the simulation process, we assumed the viscosity of GMT followed a power law as in Equation (5). The temperature-dependent effect displayed an exponential behavior as in Equation (6).

(5)$$\eta =k\;{\dot{\gamma }}^{m-1}$$(6)$$k=B\;\exp\left(\frac{T}{T_{0}}\right)$$ where k is a rheological parameter measured in (Pa·s), m is the power-law index, B is a constant (Pa·s), *T*_0_ (K) is the reference temperature.

Moreover, on the basis of mass, momentum, and energy conservations, we can derive the mathematical relationship between machine loading force and squeeze speed from Equation (7) below. Then, we will utilize this information to evaluate the power-law index (m) and the rheological parameter (k).
(7)$$F={\left(\frac{2m+1}{m}\right)}^{m}\left(\frac{2\pi kr^{m+3}}{m+3}\right)\left(\frac{{\dot{h}}^{m}}{h^{2m+1}}\right)$$ where F is the loading force, *m* is the power-law index, *r* is the shear strain, *h* is the specimen thickness (gap height, mm) at a certain time *t* (s).

### 2.2. Materials and Specimen Preparation

The original GMT material used was 102-RGR2400 (supplied by Bond-Laminates, Brilon, Germany). A 2 mm-thick sheet of GMT was used for preparing the disc specimens, as shown in Figure 2. Inside the specimen, a polyamide (PA) resin was used as the polymer matrix that was reinforced by the continuous glass mat fiber. To study the detailed micro-structure inside the GMT, we performed a thermogravimetric analysis (TGA) test to check the original fiber content. The specimen used was about 13 mg. Before performing the TGA test, the specimen was put into an oven at 100–110 °C for 10 to 30 min to remove water. After that, the dry specimen was moved into the TGA, heating from room temperature to 520 °C at 10 °C/min. The results are presented in Figure 3a. When the temperature was continuously increased, we observed 1.3 wt % decomposition at about 100 °C, corresponding to moisture loss. When the temperature reached 380 to 480 °C, the weight was reduced dramatically. This was due to the polymer matrix (PA material) decomposition (about 30 wt %) into CO_2_ and H_2_O. When the temperature was further increased from 450 °C, the weight of the specimen no longer changed. The final weight corresponded to the fiber content. Hence, the original fiber content was about 60 wt % as determined by the TGA test, as shown in Figure 3a. Moreover, to evaluate the fiber length structure, the original GMT material was weighed and then put into a furnace to perform sintering (burning) at 600 °C for 4 hours. The material exhibited internal long fiber bundles (about 54 mm length) with a random three-dimensional staggered structure, as shown as in Figure 3b.

### 2.3. Experimental Setup and Related Information

To perform compression molding, the real system with a heating chamber was setup, as shown in Figure 4a. It was modified using the Instron machine which offers the maximum loading force of 10 kN. Moreover, the key compression zone was operated in the compression mold which had an upper and a lower plate with a diameter of 50 mm. The upper mold is movable, and a controlled compression was applied at a speed of 0.001–10 mm/s. The basic operation conditions are listed in Table 1. The melt temperature was around 290–300 °C. Different compressing rates (closing speeds) ranging from 1.0 to 0.1 mm/min were applied to compress the GMT to a final thickness of 1.0 mm. The original diameter of the specimen was 25 mm. In Figure 4b, one GMT specimen was compressed between the upper and the lower molds, and an aluminum foil was used to avoid attachment of the polymer matrix to the mold surfaces.

### 2.4. Numerical Simulation Setup and Related Information

To study the viscoelastic behavior of the compression molding, a numerical simulation system was also created similar to the experimental system. Specifically, the internal compression zone and meshing setting are presented in Figure 5. For the compression molding of GMT, the process conditions were the same as those of the experiment, reported in Table 1.

## 3. Results and Discussion

### 3.1. Basic Compression Experiment and Data Analysis

First, we performed some basic compression molding tests to evaluate the viscoelastic parameters. During compression, it was assumed the no-slip boundary at the surface of the compression plate. Before compression, the testing specimen was prepared with a round shape and an original diameter of 25 mm. After compression, the real diameter of the sample was about 38 mm, as shown in Figure 6. It is clear that both the mat and the melt were deformed mainly in the lateral direction during compression. Moreover, during the compression molding testing, the relationship between the loading force (thereafter indicated as “load”) and the displacement under different compression speeds was determined. In isothermal conditions, the loading force exponentially increased with the displacement, as shown in Figure 7. Also, the load increased with the increase of the compression speed.

After performing a series of load-vs-displacement measurements from a constant initial gap and a constant temperature, the relationship between loading force and compression speed was linearized according to Equation (7), as shown in Figure 8. After data analysis at h = 1.3 mm and 290 °C, the power-law index (m = 0.6205) and the rheological parameter (k = 1.3618 × 10^6^ Pa·s) were obtained. Then, the viscoelastic parameters were implemented into the CAE software for the calculation of viscosity and the theoretical estimation of the loading force.

### 3.2. Deviations between Numerical Simulation and Experimental Study

After obtaining the viscoelastic parameters experimentally, the data were input into Moldex3D to integrate them with other parameters, including the specific volume against pressure and temperature (PVT), the heat capacity (Cp), the thermal conductivity (K), for compression molding simulation. Here, the geometrical model for the simulation was exactly the same as that of the experiment, as shown in Figure 1 and Figure 5. The diameter of the compression zone was 50 mm, and the height of the compression zone was 1.0 mm. The final thickness of the compressed charge was 1.0 mm (compressed from h_0_ = 2.0 to h = 1.0 mm). Other operation conditions are listed in Table 1. The load behavior after compression is displayed in Figure 9. The results show the relationship between load and displacement. When the compression speed was fixed, the load increased exponentially as the displacement increased. As the compression speed increased, the increase of the load at a higher speed was much faster than that at a lower speed, as shown in Figure 9a. Moreover, from the simulation results, the trend of the relationship between load and displacement was consistent with that of the experiments. However, when compared more carefully the simulation and the experiment, some significant deviations appeared, as shown in Figure 9b. Specifically, when the displacement was larger, the deviation between the predicted load of simulation and that of the experiment was larger. In addition, for safety reasons, the maximum limit of the load was set to 6000 (N) in the experiment. The real displacement was stopped at 0.7 mm (the gap was 1.3 mm), while in the simulation, it reached the final target of 1.0 mm. In addition, at the same displacement (for instance 0.7 mm), the experimental load (6000 N) was much higher than that predicted in the simulation (4000 N). Why this deviation occurred needs further study.

### 3.3. Search for the Mechanism Responsible of the Deviation

#### 3.3.1. Fiber–Polymer Matrix Separation Effect

To determine the mechanism of the deviation between simulation prediction and experiment, we analyzed the final compressed parts carefully. When the compression molding is executed, the moving upper mold tries to squeeze the GMT in the radial direction. In the simulation, we assumed that the internal fiber content is constant all the time during the compression processing. However, in real experiments, we observed that as the radial force continuously pushed the fiber–polymer matrix moving it, the polymer moved more easily than the entangled long fibers (as shown in Figure 3b). This caused the fiber and the polymer matrix to separate. To verify this fiber–polymer matrix separation phenomenon, we performed several tests as follows. In the basic testing, as described earlier, the final sample diameter was 35 mm from the original 2 mm, at the final stop of 1 mm, through theoretical calculation. On the other hand, the real diameter of the compressed sample was around 38 mm in Figure 6. The real compressed disc was 3 mm larger than the value estimated by the simulation. This was due to the separation of the polymer matrix and the fibers, which will be discussed later. The separated resin was further prompted to jump out by inertia.

Moreover, as we expected, if the polymer matrix was further pushed outward, the relative fiber-to-polymer percentage in the center portion of the specimen increased. To verify our expectation, we selected the center portion of the GMT (Figure 10a) for TGA analysis. The operation was as described in Section 2.2. Clearly, at the end of the compression, the fiber percentage in the center portion of GMT increased to 85% compared to its original value of 63%, as shown in Figure 3a. In addition, we also collected the compressed specimens subjected to three different compression speeds, and the results indicated similar fiber contents, as shown in Figure 11. Clearly, it can be stated that the fiber–polymer matrix separation effect occurring in the real experiments on continuous-fiber GMT is one of the main driving forces causing the deviation of the loading force in compression molding between the simulation prediction and the experimental results.

#### 3.3.2. Power-Law Index Estimation in Continuous-Fiber GMT 

To understand the fiber–polymer matrix separation effect in the real experiments, we extended our investigation to the power-law index variation during compression molding. Using Equation (7), we focused on different compressed heights and estimated the power-law index. Surprisingly, the internal fiber microstructure changes influenced the power-law index to change significantly, as shown in Figure 12. As we described earlier, as compression proceeds, the squeeze force pushes outward the polymer more than the continuous long fibers, as a consequence of fiber entanglement. This causes the relative fiber-to-polymer percentage in the internal portion of the specimen to increase gradually. This is further reflected by changes in viscoelasticity, from the start (h0 = 2.0 mm) to the end (h = 1.0 mm) of compression molding.

#### 3.3.3. Power-Law Index Estimation in Pure PA 

Furthermore, to validate the fiber–polymer matrix separation phenomena in the real experiments, we also applied pure PA (Altech PA66 ECO 1000/561 Black BK0002-00, supplied by ALBIS company, Hamburg, Germany) and measured the associate viscoelastic parameters. The operation conditions and specimen dimension are listed in Table 2. Since there was no fiber entanglement effect, the compression behavior was much simpler than that of GMT. Therefore, the operation parameters were different. The relationship between load and displacement, after performing compression molding at three different compression speeds experimentally, is shown in Figure 13. In isothermal conditions, the load was proportional to the displacement. Besides, the load increased with the increase of the compression speed. These results are similar to those of the GMT system.

After performing a series of load vs. displacement measurements from a constant initial gap and at a constant temperature, the relationship between load and compression speed was linearized according to Equation (7). After data analysis, the power-law index (m = 0.8) and the rheological parameter (k = 430 Pa·s) was estimated. After obtaining the viscoelastic parameters experimentally, as mentioned earlier for the GMT system, the data were implemented into Moldex3D for compression molding simulation, as described in Section 3.2. Here, the diameter of the compression zone was 50 mm, and the height of the compression zone was 2.0 mm. The final thickness of the compressed part was 1.0 mm. Other operation conditions are listed in Table 2. The load behavior after compression is displayed in Figure 14. The results show that when the compression speed was fixed, the load increased exponentially as the displacement increased. As the compression speed increased, the increase of the load at a higher speed was much faster than that at a lower speed. Moreover, from the simulation results, the trend of the relationship between load and displacement was consistent with that of the experiments. Compared to the GMT system, the pure PA system did not display a significant deviation between the numerical simulation and the experimental study. Moreover, using Equation (7), we focused on different compressed heights and estimated the power-law index. The comparison of the power-law index of the GMT and pure PA systems is presented in Figure 15. Here, several features need to be addressed. First, since there is no fiber inside the pure PA system, the flow behavior of the pure polymer was smoother. The PA system had a constant power-law index with m = 0.8. In addition, the curve-fitting rheological parameter of pure PA was also close to a constant value of 460 Pa·s. In contrast, the power-law index of GMT varied (m = 1 to 0.62), and the rheological parameter also changed (k = 4.14 × 10^6^ to 1.3618 × 10^6^ Pa·s) during compression. This confirmed that the fiber–polymer matrix interaction was significant in the GMT system in compression molding.

## 4. Conclusions

In this study, we propose a method to measure the rheological properties of GMT through a compression system. Specifically, we utilized compression molding to record the relationship between loading force and displacement. This quantitative information was used to estimate the power-law index and viscoelastic parameters and for viscosity prediction. Moreover, the estimated viscoelastic parameters of GMT were implemented into Moldex3D to evaluate the flow behavior under compression. The simulation prediction and experimental measurements were further compared. The results showed that the loading force increased exponentially against the displacement at various compression speeds. The trend of the loading force variation was consistent for both numerical simulation and experiments. However, there were some significant deviations. Specifically, at the end stage of compression molding, the experimental loading force was much higher than that estimated by the simulation. The mechanism causing this deviation was expected to be related to the fiber–polymer matrix separation effect. To validate our expectation, we have applied TGA to analyze the fiber content of the center portion of the compressed specimens. The fiber content increased from 63% to 85% during compression. This fiber content variation influences the power-law index and the rheological parameters of GMT making them fluctuate. Specifically, the power-law index changed from 1.0 to 0.62. These internal rheological properties changes further increased the loading force in the real experimental GMT system. To further verify the change in the rheological properties, pure PA was used to perform compression molding tests. The results showed that because of the absence of fiber–polymer matrix interactions in the pure polymer PA system, the power-law index and the curve-fitting rheological parameters were almost constant. The mechanism causing the deviation was therefore validated.

## Figures and Tables

**Figure 1 polymers-11-00335-f001:**
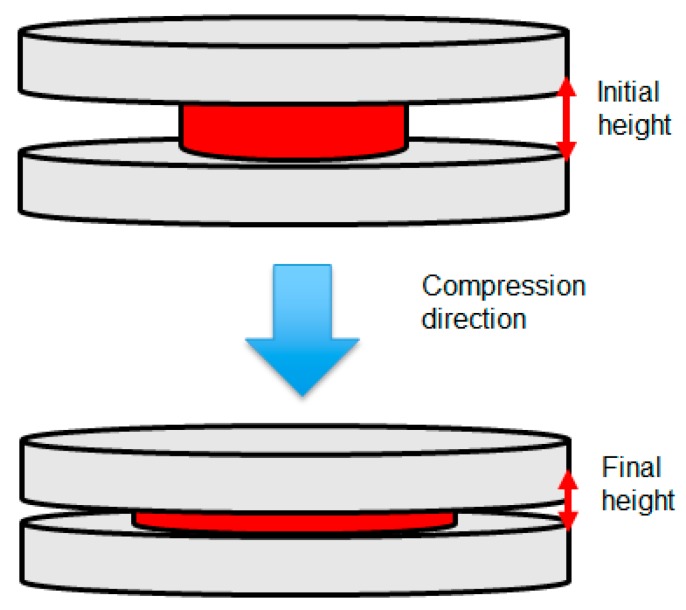
Schematic of the compression system.

**Figure 2 polymers-11-00335-f002:**
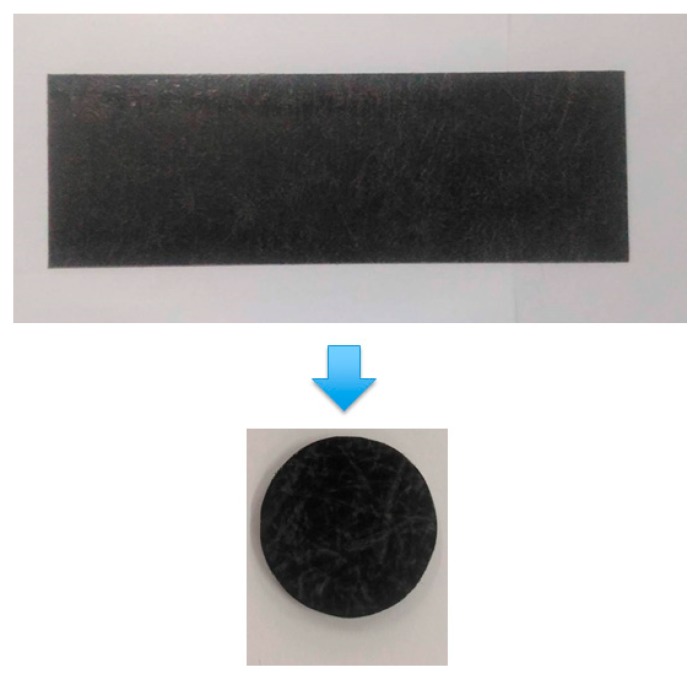
Glass fiber mat thermoplastics (GMT) specimen preparation: making a 2 mm-thick round disc with a diameter of 25 mm.

**Figure 3 polymers-11-00335-f003:**
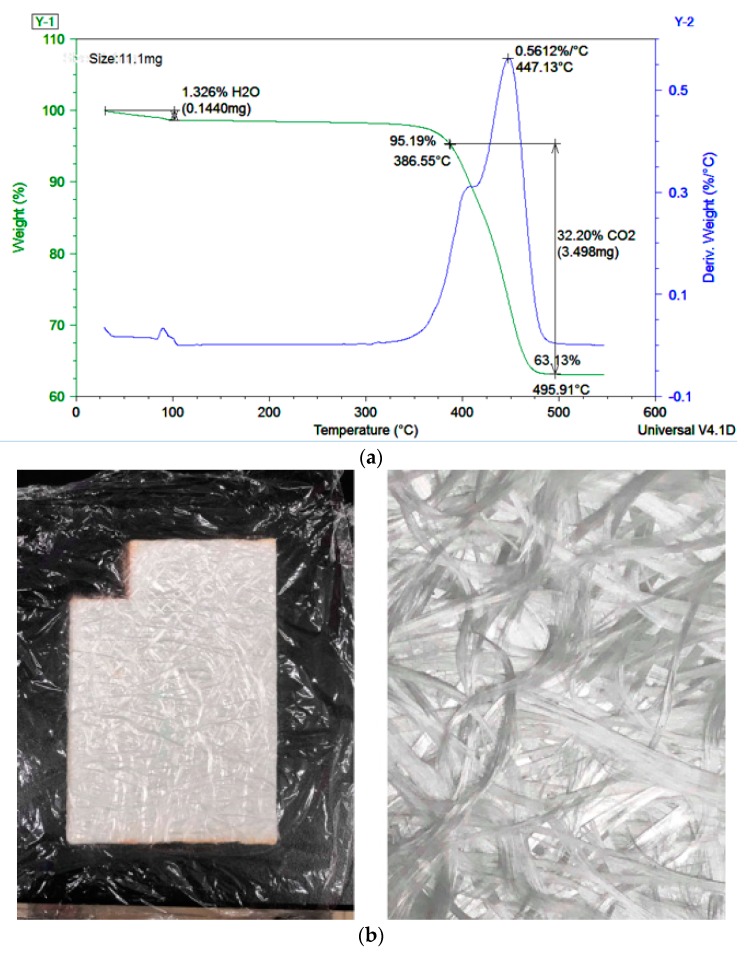
Characterization of GMT micro-structure before the compression molding test: (**a**) TGA analysis result, (**b**) after the burning testing.

**Figure 4 polymers-11-00335-f004:**
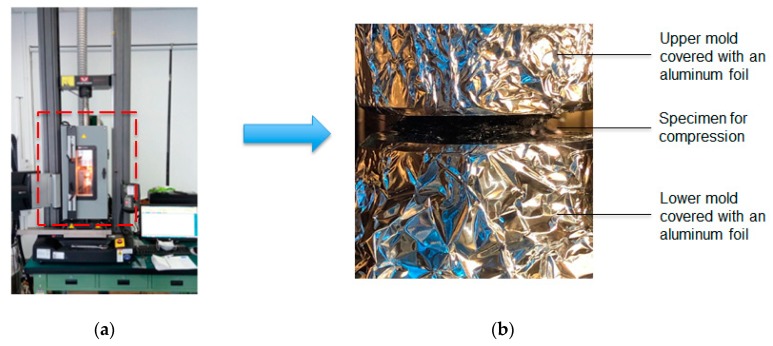
Compression molding system: (**a**) heating chamber, (**b**) compressed specimen inside the compression mold in an Instron machine.

**Figure 5 polymers-11-00335-f005:**
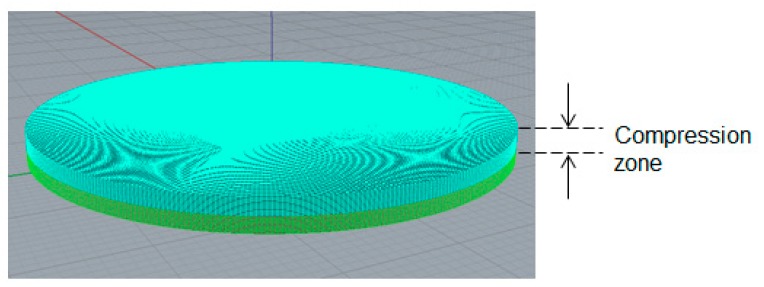
Schematic of the numerical compression system, showing the compression zone.

**Figure 6 polymers-11-00335-f006:**
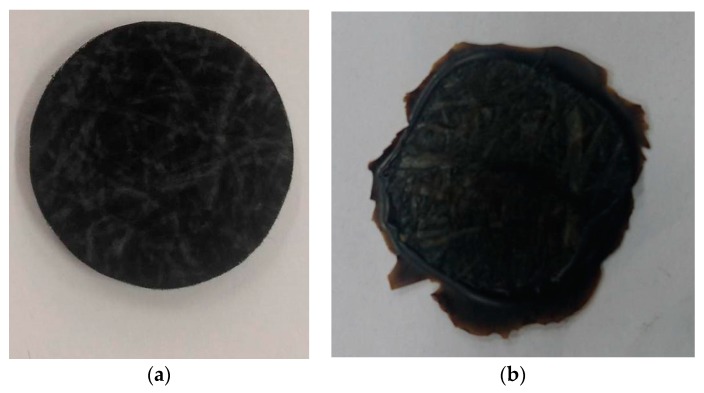
Shape of the GMT specimen: (**a**) before compression: diameter = 25 mm, (**b**) after compression: diameter = 38 mm.

**Figure 7 polymers-11-00335-f007:**
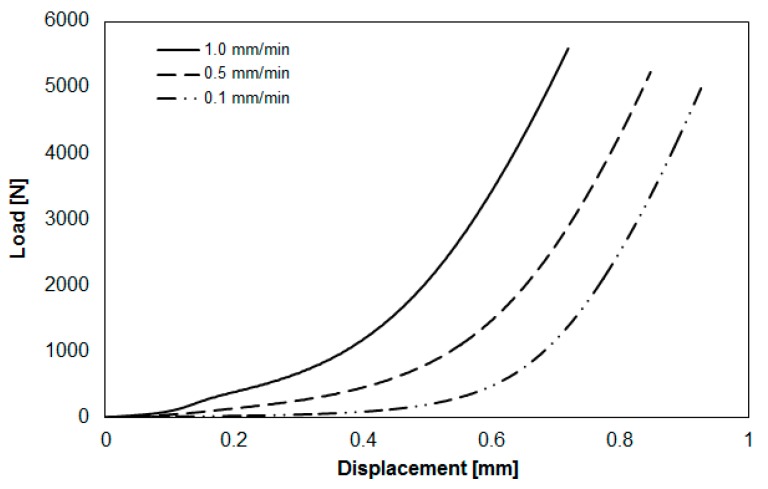
Loading force vs. plate displacement at 290 °C and compression speed of 0.1 to1.0 mm/min.

**Figure 8 polymers-11-00335-f008:**
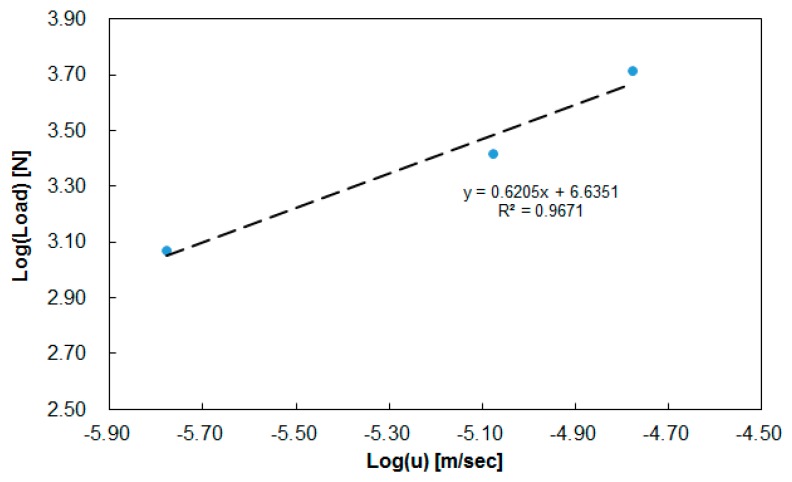
Linearization of the relationship between loading force and compression speed at a constant gap of 1.3 mm and a temperature of 290 °C.

**Figure 9 polymers-11-00335-f009:**
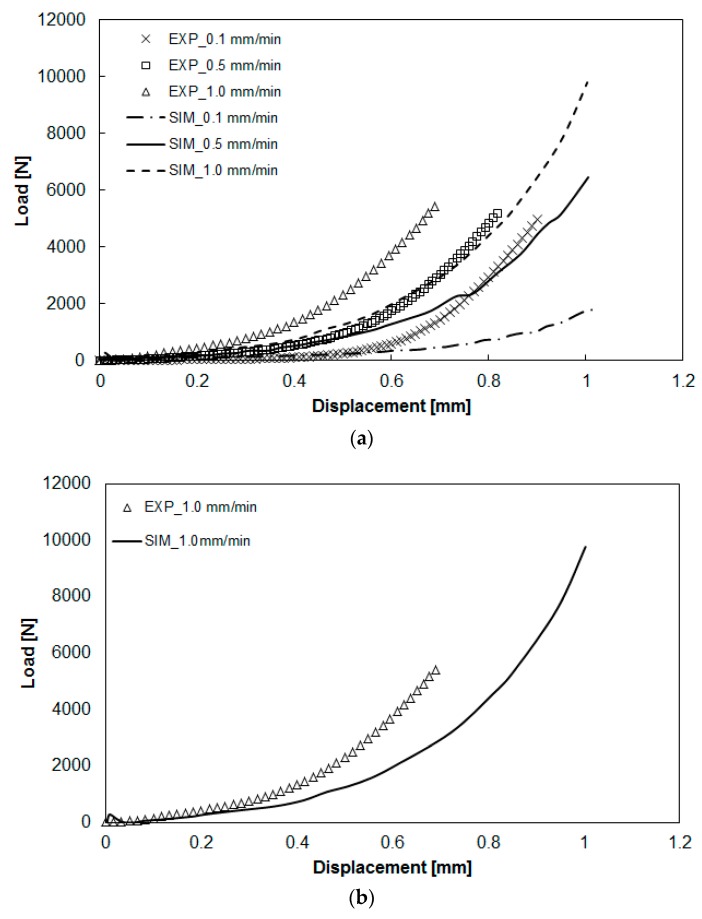
Comparison of the load-vs-displacement relationship between simulation and experiment: (**a**) effect of different compression speeds, (**b**) some significant deviation areas observed between simulation and experiment.

**Figure 10 polymers-11-00335-f010:**
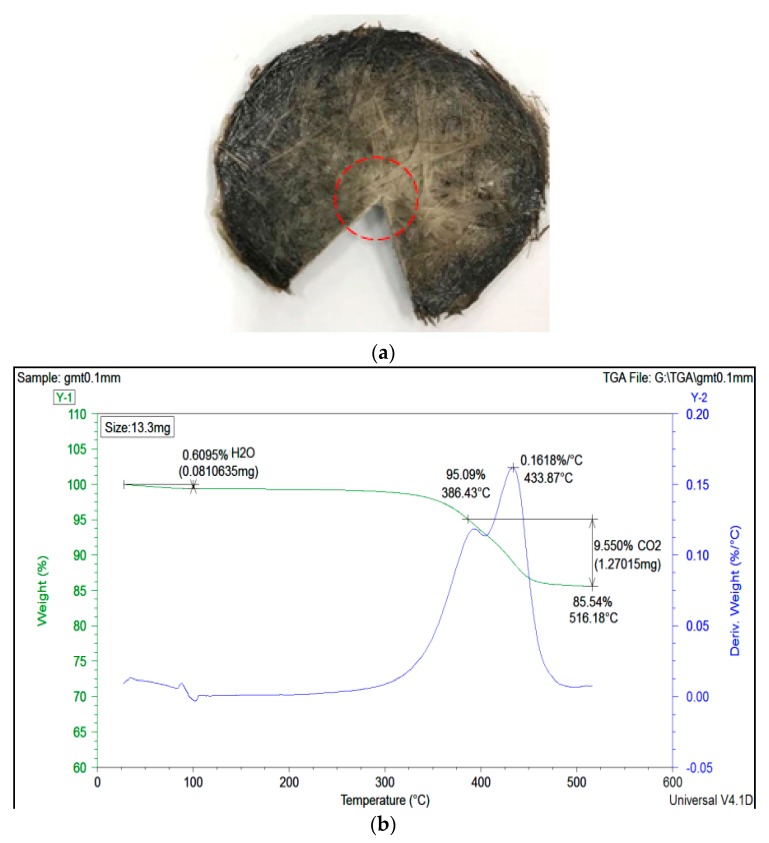
(**a**) Final, compressed GMT, the red circle shows the portion used for TGA measurements, (**b**) TGA curve of the compressed GMT area.

**Figure 11 polymers-11-00335-f011:**
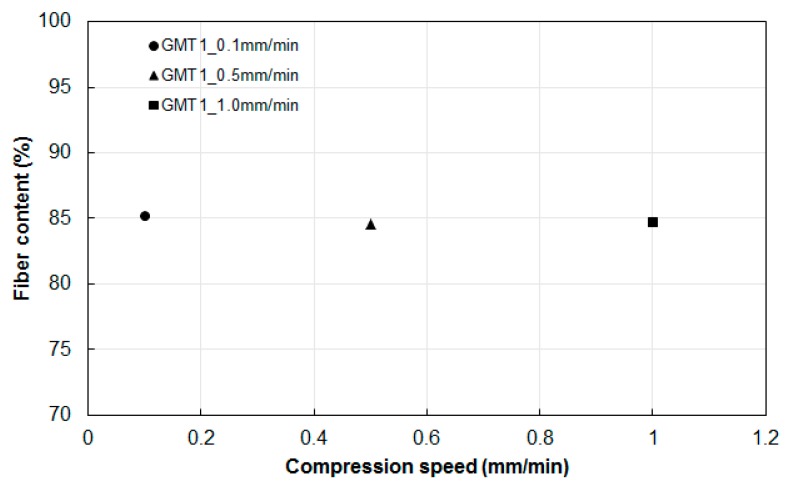
Fiber concentration measurement of the compressed products at different compression speeds.

**Figure 12 polymers-11-00335-f012:**
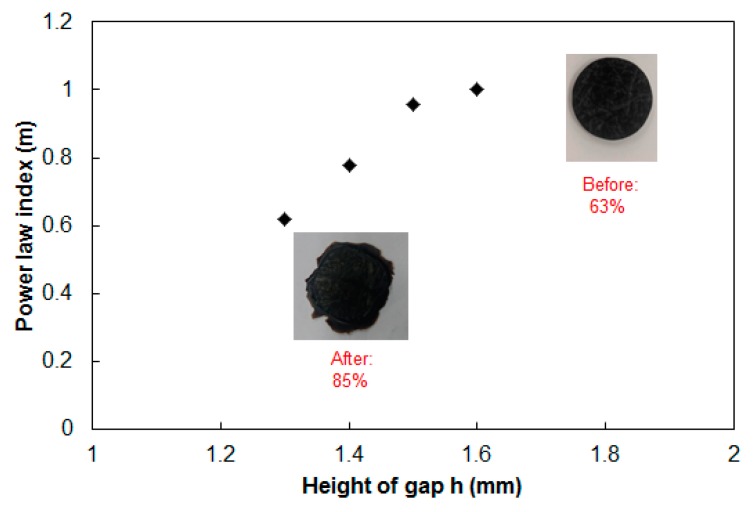
Power law-index of the testing material GMT1 during compression.

**Figure 13 polymers-11-00335-f013:**
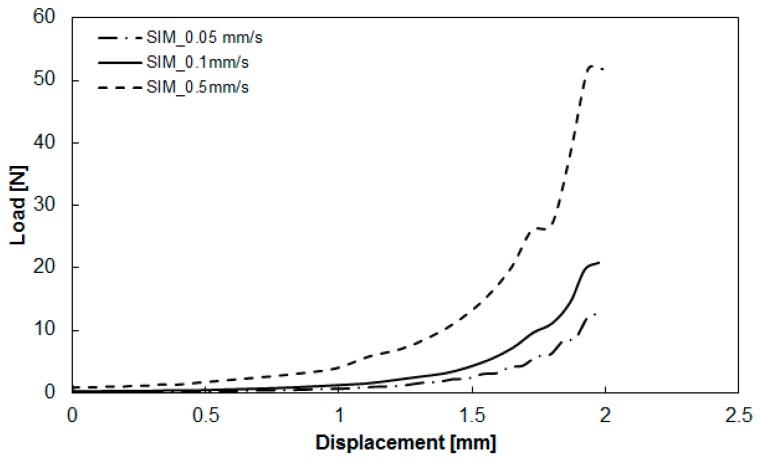
The loading force vs. plate displacement at 290 °C and a compression speed of 0.1 to 1.5 mm/s.

**Figure 14 polymers-11-00335-f014:**
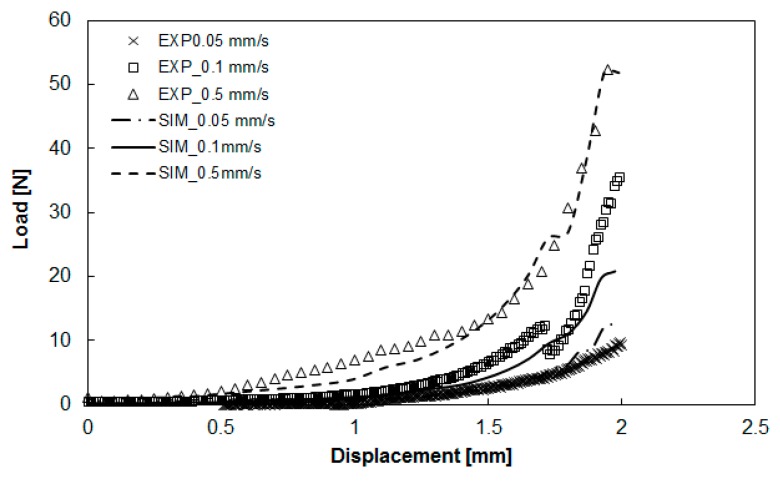
Comparison of the loading vs. displacement between simulation and experiments: effect of different compression speeds.

**Figure 15 polymers-11-00335-f015:**
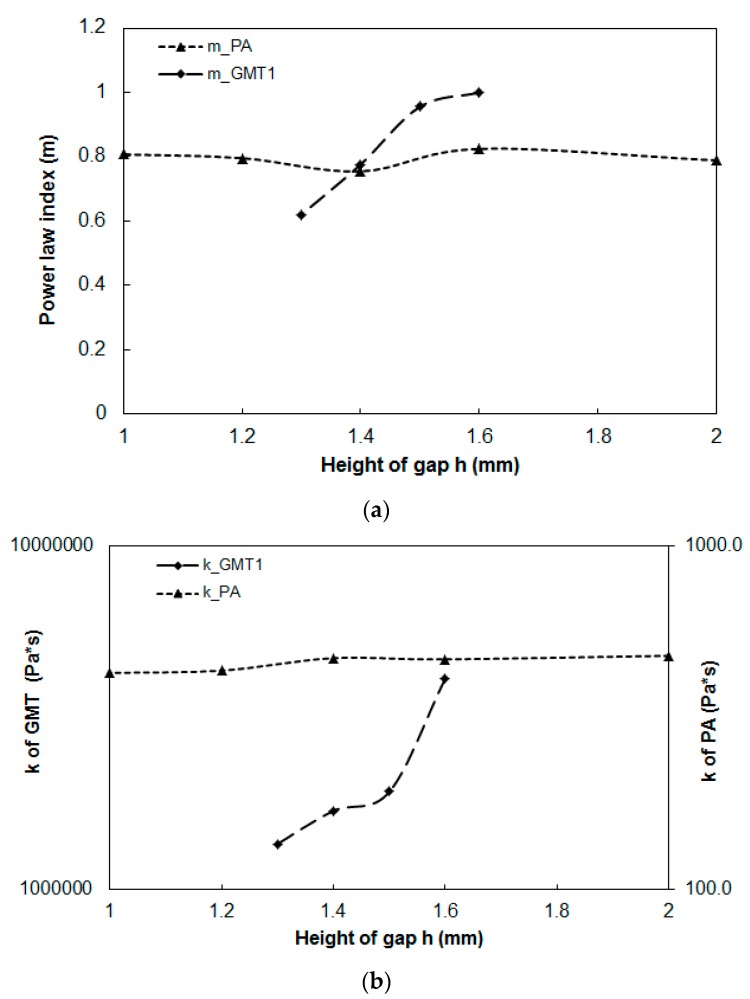
Comparison of the rheological parameters for the GMT and the pure PA systems: (**a**) power-law index, (**b**) k value.

**Table 1 polymers-11-00335-t001:** Operation conditions for compression molding of GMT.

**Temperature [°C]**	290−300
**Compression Speed [mm/min]**	0.1−1.0
**Initial Location [mm]**	2
**Final Location [mm]**	1
**Sample Diameter [mm]**	25
**Sample Thickness [mm]**	2

**Table 2 polymers-11-00335-t002:** Operation conditions for compression molding for pure polyamide (PA).

**Temperature [°C]**	280–300
**Compression Speed [mm/s]**	0.1–1.5
**Initial Location [mm]**	3
**Final Location [mm]**	1
**Sample Diameter [mm]**	25
**Sample Thickness [mm]**	3

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
