# Peer review of "Investigation of the Viscoelastic Behavior Variation of Glass Mat Thermoplastics (GMT) in Compression Molding"

_polymers, 2019, doi:10.3390/polym11020335_

Round 1
Reviewer 1 Report
In my first look to the paper, my decision was to reject it. The introduction mixes compression molding and injection molding in a way that does not serve the interest of the manuscript and confuses the reader, the manuscript is to long for the work that is presented, some of the statements have lack of scientific sound (eg. "It would be water."), and for most, the experimental setup is very academic and its scale up for a real part is not feasible. Nevertheless, in a second read, I believe that there is information on the paper that may be useful to the scientific community, therefore I recommend that the paper is accepted after a revision of the manuscript to increase its soundness. This included also the figures. Most of them have low quality and are not presented in a style coherent with the manuscript style (eg. big size words in figures, comparable to the manuscript text size)
Author Response
Our response:
Thank you very much for your patience and priceless suggestions.
The Introduction Section has been revised.
To avoid confusion, some portion with injection molding is deleted. (Page 1, Line 39) “when the fiber reinforced thermoplastics goes through injection molding processing, the length of the fiber is broken dramatically. Hence,” was deleted.
(Page 1, Line 43) “switching to” is changed to “using”
(1) To improve the scientific soundness, several portions are modified as follows.
(Page 4, Line 126) “original” is deleted.
(Page 4, Line 128) “expected” is changed as “used”.
(Page 4, Line 133) “100- 520oC” is modified as “room temperature to 520oC”.
(Page 4, Line 134-135) “It would be water.” is revised as “That is the amount of moisture”.
(Page 4, Line 137) “It is expected that” is deleted.
(Page 4, Line 138) “Hence,” is added.
(Page 4, Line 139-140) “the original GMT material has been weighed before to send them into the furnace. Then put the specimen into furnace …” is modified as “the original GMT material has been weighed and then put into furnace …”.
(Page 5, Line 165) “with a heating chamber” is added.
(Page 5, Line 167) “To provide isothermal environment, it has a heating chamber for temperature control.” is deleted.
(Page 5, Line 169) “Moreover” is revised as “Furthermore”.
(Page 5, Line 172-174) “In Figure 4(b), one GMT specimen is compressed between upper and lower molds, where aluminum foil is used to avoid polymer matrix to stick on the mold surfaces.” is added.
(Page 6, Line 190) “are same that” is modified as “are the same as that”.
(Page 7, Line 215) “estimated” is revised as “obtained”.
(Page 8, Line 234) “could be” is changed as “was”.
(Page 8, Line 249) “loading force” is modified as “load”.
(Page 8, Line 252) “loading force” is revised as “load”.
(Page 9, Line 265) “loading” is changed as “load”.
(Page 10, Line 282) “However” is modified as “On the other hand”.
(Page 10, Line 283) “It is clear that both the mat and the melt were deformed mainly in lateral direction during compression. However,” is deleted.
(Page 10, Line 283) “It might be” is changed as “It is”.
(Page 10, Line 289) “in Section 2.2” is added.
(Page 10, Line 289) “The dry specimen was moved into TGA heating from 100-520oC at 10oC/min. When the temperature is continuous increased, it shows 0.6 wt% decomposed at around 100oC which is water. When it is touched 380oC to 480oC, the weight is reduced dramatically. It is due to the polymer matrix (PA material) decomposed (about 9.5 wt%) into CO2 and H2O. When the temperature is kept increased from 450oC and higher, the weight of specimen is no changed. It is expected that the final weight is the fiber content.” is deleted.
(Page 12, Line 351) “Due to” is revised as “Since”.
(Page 12, Line 353) “a little bit” is deleted.
(Page 12, Line 362) “could be input” is modified as “was implemented”.
(Page 12, Line 362) “to integrate with other parameters, including the specific volume against pressure and temperature (PVT), the heat capacity (Cp), the thermal conductivity (K),” is deleted.
(Page 12, Line 363) “as described in Section 3.2” is added.
(Page 12, Line 375) “might be” is modified as “is”.
(Page 12, Line 375) “It is expected that” is changed as “The”.
(2) Most of Figures have been modified with better quality and suitable text size in the figures.
Figure 1 is revised.
Figure 3(a) is modified.
Figure 4(b) is updated.
Figure 5 is changed.
Figure 7 is revised.
Figure 8 is modified.
Figure 9(a) and (b) are revised.
Figure 11 is changed.
Figure 12 is modified.
Figure 13 is revised.
Figure 14 is changed.
Figure 15(a) and (b) are revised.

Reviewer 2 Report
This manuscript presents an experimental and numerical study on viscoelasticity behavior of GMT during compression molding. The strain rate and loading force was controlled and measured by an Instron machine with result fitting to obtain power law index and rheological parameter. Suitable content is provided which is of interest to the readers of Polymers. This reviewer believes minor revisions are required before being accepted for publication.
1. Page 6, figure 4(B). Larger picture showing only the plate and sample should be better to display relative size of plate and sample. Is this a picture of sample after compression? If so, please note in the caption.
2. Another assumption may need is that the expansion at radius direction is equal across the sample to ensure uniform thickness and loading force.
Author Response
Page 6, figure 4(B). Larger picture showing only the plate and sample should be better to display relative size of plate and sample. Is this a picture of sample after compression? If so, please note in the caption.
Response 1:
Thank you very much for the indication.
Regarding to Figure 4(b), it showed the procedure of inserting GMT sample into compression mold. I think your suggestion is great, to get better understanding, the new picture 4(b) with a compressed specimen has been updated as below.
The caption is also modified.
(Page 5, Line 172-174) “In Figure 4(b), one GMT specimen is compressed between upper and lower molds, where aluminum foil is used to avoid polymer matrix to stick on the mold surfaces.” is added.
Figure 4. The compression molding system: (a) the heating chamber, (b) the compressed specimen inside the compression mold in an Instron machine.
2. Another assumption may need is that the expansion at radius direction is equal across the sample to ensure uniform thickness and loading force.
Response 2: Thank you very much for the great suggestion. The assumption statement is added in Page 3 (Line 100-101).
“In addition, it is assumed that the expansion at radius direction is equal across the sample to ensure uniform thickness and loading force.”
